# A Nursing Process for Shared Decision-Making for Patients with Severe Mental Illness Receiving Treatment Involving Long-Term Coercive Measures: A Modified Grounded Theory Approach

**DOI:** 10.3390/healthcare12100967

**Published:** 2024-05-08

**Authors:** Yutaka Nagayama

**Affiliations:** 1School of Nursing, Kanazawa Medical University, 1-1 Uchinada, Kahoku 920-0265, Ishikawa, Japan; naga-y@kanazawa-med.ac.jp; Tel.: +81-76-218-8425; 2Nursing Department, Kanazawa Medical University Hospital, 1-1 Uchinada, Kahoku 920-0265, Ishikawa, Japan

**Keywords:** shared decision-making, severe mental illness, coercive measures, modified grounded theory approach

## Abstract

The use of shared decision-making (SDM) has recently attracted attention for building recovery-oriented therapeutic relationships with patients with severe mental illness (SMI). The purpose of this study was to describe a nursing process for SDM for psychiatric patients with SMI being treated via long-term coercive measures, such as seclusion and physical restraint, in the “seclusion room” of a psychiatric ward. The study used a modified grounded theory approach. The participants were 17 psychiatric nurses. Data were collected via semi-structured interviews. Concepts and categories were generated from verbatim transcripts, and their relationships were illustrated using a diagram and by generating a storyline. The nursing process for SDM was based on sensing the response to triggering stimuli, and the nurse-led preventive measures compensated for the patients’ lack of coping skills. Because of the patients’ persistent instability in response to certain stimuli, in our process, nurses are involved in creating opportunities for self-understanding and self-selection while also taking proactive preventative measures. Patients’ reactions to surrounding stimuli were evaluated by nurses, who then determined whether they (or the patient) should take the lead in terms of decision-making.

## 1. Introduction

In shared decision-making (SDM), patients consider available treatment options and make informed medical choices that are favorable to them [1]. The essential elements of SDM include the involvement of both healthcare providers and patients, the sharing of information, and taking steps to reach agreement on the most desirable treatment for both parties [2]. SDM is also relevant to the field of psychiatry. The concept of SDM in psychiatric clinical practice is, in turn, related to concepts such as autonomy and person-centeredness [3]. A conceptual analysis of SDM for patients with severe mental illness (SMI) has been conducted [4]. The analysis showed that the main attributes of SDM are the relationship between patients and experts, the communication process, visualizations that are easy to understand for all parties, and a broad stakeholder approach (interprofessional collaboration and potential caregiver involvement). Furthermore, the communication process was the most complex aspect of SDM, comprising five stages: goal sharing, information sharing, reflection, mutual agreement, and follow-up.

Coercive measures, such as seclusion and physical restraint, can only be used if the psychiatric symptoms of a patient admitted to a psychiatric ward have worsened significantly such that the patient may pose a danger to themself or others. In Japan, the Ministry of Health, Labour and Welfare has established basic policies for seclusion and physical restraint, as well as the conditions that should be met for their application to patients, on the basis of the Mental Health and Mentally Disabled Persons Welfare Act [5]. Patients eligible for seclusion include those who cannot be accommodated in general wards because of suicide attempts, self-harm, violent behavior, nuisance behavior, property damage, restlessness, hyperactivity, or explosiveness. Patients subject to physical restraint are those who are at a high risk of attempting suicide or self-harming, or are hyperactive, restless, or have a mental illness that, if left untreated, can cause a life-threatening situation. Seclusion and physical restraint take place in a room in the psychiatric ward called the “seclusion room”. In seclusion rooms, patients are secluded and physically restrained so that they can be provided with medical care and protected from situations where they may harm themselves or others.

In Japan, compared with other countries, the duration of coercive measures such as segregation and physical restraint tends to be longer. In a study of inpatients who underwent seclusion or physical restraint in six acute wards of four psychiatric hospitals in Japan, the median seclusion time was 204 h, and the median physical restraint time was 82 h [6]. In a study on the relationship between coercive measures and staffing in psychiatric inpatient facilities in Japan, the odds ratio for seclusion and physical restraint increased from 1.74 to 2.36 as the number of nurses per 10 beds increased. Inpatients who require seclusion and physical restraint are typically admitted to wards with high nurse-to-bed ratios, such as psychiatric emergency wards, wherein inpatients are most likely to experience seclusion and physical restraint [7]. In a survey evaluating the characteristics of patients undergoing coercive measures in Japan, it was found that patients with schizophrenia are isolated for longer periods than patients with other diseases, and that the main reason for seclusion and restraint is to prevent harm to others [6]. Especially in patients with chronic schizophrenia, it is difficult to predict maladaptive behaviors, such as violence and verbal abuse, which are caused by cognitive dysfunction [8] and may lead to prolonged coercive measures.

Long-term use of seclusion and physical restraint increases patient suffering and can be experienced as physically, psychologically, and socially invasive. Moreover, people with SMI often experience trauma when coercive measures, such as seclusion and physical restraints, are applied in psychiatric care [9]. In one study comparing groups of people with SMI with and without a history of violence, global mental function improved over time in both groups, and there was no decrease in the risk impact on recovery. Additionally, individuals with a history of violence had fewer psychiatric symptoms before and during the survey compared with those without a history of violence [10]. Whether psychiatric symptoms should be considered as a risk factor for patients with SMI exhibiting aggressive behavior, such as violence and verbal abuse, is controversial. The use of seclusion and physical restraint to contain verbal abuse and violence should be limited to emergency situations as a safety measure because, with long-term use, the disadvantages of coercive measures as a treatment option increase for patients.

Nurses with a recovery orientation, and those working in open wards, had negative attitudes toward coercive measures [11]. In Japan, to guide people with SMI who have been secluded for long periods toward release, psychiatric nurses may seek to understand the interpersonal world of these patients by assuming their viewpoint. Nurses may communicate interactively with patients at times when the latter are not experiencing severe pathology, as well as actively create opportunities for SMI patients to pursue self-determination and ultimately improve their tolerance of stressful stimuli [12]. Thus, many psychiatric nurses seek to create environments that promote recovery from SMI, in which maladaptive behaviors such as violence and verbal abuse are suppressed. In particular, there may be a need to create care environments that do not rely on coercive measures.

The use of SDM has recently attracted attention as a means to build recovery-oriented therapeutic relationships with patients with SMI. By creating a plan with patients that is based on SDM and setting achievable goals through dialogue, mental health nurses were reportedly able to build trust and identify patients’ wishes [13]. In addition, when nurses in Japan introduced interventions that were based on the strength model for SMI patients on whom coercive measures had been applied for a long period of time, the seclusion time was reportedly reduced in half of the cases [14]. It has been suggested that the use of strength models has the potential to promote self-insight in patients, although consideration must be given to the patient’s stimulus tolerance. Promoting autonomously motivated behavior and leveraging opportunities for self-determination are encouraged to promote SMI recovery [15]. The use of SDM could help patients heal from traumatic experiences, restore self-esteem, increase autonomy, and promote adaptive behavior in psychiatric settings.

The purpose of this study is to describe the process of building a relationship in which psychiatric nurses jointly make decisions with, and solicit adaptive behavior from, patients with SMI, with the aim of minimizing the need for coercive measures. Researchers have determined that the modified grounded theory approach (M-GTA) [16], which is oriented toward theory generation and the practical application of findings even for clinical research of limited scope, is useful for describing the process of change in research participants. The M-GTA aims to explain the process of change in human behavior on the basis of social interaction. In addition, the M-GTA retains the basic characteristics of the grounded theory approach of Glazer and Strauss [17], including a focus on theory generation, a grounding in data analysis, empirical demonstrability, and application of the results. In this study, from the perspective of a psychiatric nurse, the initial goal was to generate a theory about the structure of change in interactions with patients with SMI to promote joint decision-making. The overall purpose of this study was to describe a nursing process promoting SDM for psychiatric patients with SMI receiving treatment involving long-term coercive measures, such as seclusion and physical restraint, in the “seclusion room” of a psychiatric ward.

## 2. Materials and Methods

### 2.1. Research Design

This study was conducted on the basis of the M-GTA [18]. As stated above, the M-GTA is a research method that retains and also develops the theoretical principles of the GTA of Glaser and Strauss [17]. The M-GTA aims to generate theories that can explain and predict change in human behavior on the basis of social interactions. Moreover, theories generated via the M-GTA can be generalized even on the basis of research with a limited scope.

The M-GTA aims to integrate Glaser’s consistent epistemological position (objectivism) with another position, namely constructivism, which essentially constitutes Strauss’s position. When generating a theory with the aim of generalizing, despite a limited scope of research, it is necessary to generate concepts on the basis of the perspective of the analytically target person. Theory based on a human perspective with an analytical focus may be more applicable for practitioners in similar fields. This study targets the phenomenon whereby psychiatric nurses collaborate with patients with SMI, through their social interactions, to help the patients acquire adaptive behavior in the care environment to minimize the need for coercive measures. A theory of the structure of SDM in nursing practice for patients with SMI, from the perspective of psychiatric nurses, may be suitable for practical application in similar clinical settings. On the basis of the characteristics of the M-GTA described above, it was determined that this approach is suitable for the purpose of this study.

### 2.2. Participants

The participants were nurses working in psychiatric wards at six hospitals. The inclusion criteria applied in this study were as follows: working in the current ward for >1 year; currently working full-time and during regular working hours; and having experience in caring for patients whose treatments included being secluded or physically restrained for >1 month, or on multiple different occasions during their hospitalization period.

### 2.3. Data Collection

An interview survey was conducted with each individual participant. The interview time was approximately 30 min. Interviews were conducted in a face-to-face or online setting. The researcher asked the nurses about the number of years that they had been working as a nurse, including in a psychiatric setting. The interview covered situations involving decision-making during interactions with patients in the seclusion room, including decisions pertaining to daily life, nursing plans and content of care, and releasing patients from their seclusion and physical restraints. The interviews were recorded using a voice recorder and transcribed verbatim.

### 2.4. Data Analysis

In the M-GTA, an analytical theme and an “analytically-focused person” (i.e., the focus of the research) are identified. In this study, the analytically focused persons are “nurses who provide nursing care to patients whose treatment includes long-term coercive measures implemented in psychiatric wards”, and the analytical theme is “the process of collaboratively building patients’ coping skills and improving their living environment to minimize the requirement for use of seclusion and physical restraint”. Concepts were identified through interpretation of the data by utilizing the analytical theme and analytically focused person. Once a concept had been generated, its semantic content was refined by applying the method of continuous comparative analysis, in which data were checked in light of a similar meaning and opposite meaning. Once multiple concepts had been generated, the researcher examined the relationships between them. Finally, the researcher created a diagram and generated a storyline to illustrate the relationships between concepts and categories.

### 2.5. Ethical Considerations

This study was conducted with approval (Approval Number: I657) from the Medical Research Ethics Review Committee of Kanazawa Medical University. We explained the research plan, verbally and in writing, to the inpatients, who were invited to participate and provided consent. Participation in this research was voluntary, and individuals who declined to participate did not experience any disadvantages in relation to that decision. Personal information was anonymized and coded.

## 3. Results

### 3.1. Participants’ Demographic Information

In total, we recruited 17 participants. The average number of years of nursing experience was 16.2, and the average number of years of psychiatric nursing experience was 11.4. The participants’ demographic information and patients’ characteristics, as described by the participants, are shown in Table 1.

### 3.2. Storyline

Four categories and nine concepts were generated. Figure 1 diagrammatically shows the relationships between concepts and categories. The storyline illustrating the relationships between the concepts is shown in Figure 1 below.

Nurses aim to achieve an understanding of their patients’ thoughts and feelings by “connecting with their fragmented thoughts and feelings”. “Sensing the response to triggering stimuli” refers to nurses’ assessments of the degree of restlessness or agitation in a patient. On the basis of this concept, nurses take “nurse-led preventive measures”, such as “do not force it out” (in the context of the patient’s thoughts and intentions), and they focus on “creating an environment where the patient can remain calm”.

By “creating an environment where the patient can remain calm”, space is made in which dialogues can be established, and by “repeatedly asking about progress and prospects” and focusing on “responses that lead to realistic understanding”, nurses can be considered to be “promoting self-awareness of maladaptive behavior patterns”. At the same time, nurses make “suggestions for actionable preventive measures”, such as “negotiating means to satisfy goals and wishes” and “negotiating acceptable arrangements that suit the patient”. On the basis of these negotiations, the nurses were involved in “creating a place where patients can be motivated to be active” and helped promote “self-choice in activities and symptom management”.

Because of patients’ persistent instability in response to certain stimuli, nurses aim to create opportunities for self-understanding and self-selection, while also taking proactive preventative measures.

### 3.3. Categories and Concepts

The meanings and specific examples of categories and concepts are provided below. Sections in italics indicate the participant’s narrative.

#### 3.3.1. Category: Sensing the Response to Triggering Stimuli

Nurses determine the extent to which patients respond to triggering stimuli that cause restlessness or agitation. The sensitivity of the patient influences the likelihood of behavior that warrants the use of a seclusion room. The nurse carefully observes how the patient’s sensitivity affects their subsequent response. For example, when a patient’s desires are thwarted by a medical professional, or when a patient is forced into a situation where they are excessively particular or inflexible, nurses can sense whether a patient has become restless or agitated. In addition, even during open observation, patients are observed from a distance to ensure that they do not cause trouble with other patients. Nurses are well aware that the patient’s sensitivity plays a role in the likelihood of behavior that warrants the use of a seclusion room. Nurses always carefully monitor patients for changes in their mental state, and for impulsive behavior, associated with stimuli that could trigger restlessness, excitement, or violence. As a prerequisite for joint decision-making, observation is needed at every step, i.e., it is necessary to check patients’ responses to environmental stimuli.

##### Concept: Connecting with Their Fragmented Thoughts and Feelings

Nurses piece together patients’ thoughts and feelings, which may be expressed in fragments, in a way that is easy for both the patient and the nurse to understand. Patients find it extremely difficult to express their feelings and thoughts verbally, and it is similarly difficult for them to perceive and share the feelings that are arising inside them. For this reason, nurses closely observe patients to determine the extent to which they are able to verbally express their emotions. Nurses try to “bring out” the emotions that patients may be holding within themselves as much as possible. Furthermore, nurses try to compensate for any deficits in patients’ capacity for emotional expression and help them organize their thoughts so that they can express what they really want. Through such interventions, nurses help patients understand their true desires.


*When I ask patients, “Why do you think you’re feeling well now?*
*” they say they don’t know. This shows that patients have a hard time expressing their feelings in their own words. When a nurse says, “How about this?*
*” or “This is what it is, so this is what it is”, the patient’s answers often seem like those of parrots. Nevertheless, the patient can answer.*
(J)

#### 3.3.2. Category: Nurse-Led Preventive Measures

Nurses apply nursing-led coping skills to prevent the worsening of mental symptoms. Patients may be unable to recognize or deal with signs of deteriorating mental status on their own. Patients find it difficult to self-monitor as part of their self-care. Therefore, the nurse will tell the patient when a “red flag” is present. For example, when a patient is in a good mood, they may be unable to control their emotions. In such situations, nurses try to avoid coercive measures by encouraging patients to participate in activities that interest them, and sometimes by calming them down by using medications as needed. Therefore, the nurse tells the patient when the next warning signs that they should be aware of are likely to occur. A pattern has been identified in which patients find it difficult to deal with their mental symptoms themselves and rely on the assistance of nurses. In other words, nurses supplement patients’ self-monitoring efforts.

##### Concept: Do Not Force It Out

If the nurse’s suggestion is not acceptable to the patient, the patient’s natural inclinations should be respected without forcing them to disclose their thoughts or make suggestions. When it is determined that the nurse’s advice and suggestions are not acceptable to the patient, it may be that the patient is concerned that the nurse’s intervention will be invasive such that their mental state will further deteriorate. Furthermore, in some cases it was determined that the nurses’ administrative actions were not in the best interest of the patient. For example, to cope with auditory hallucinations, one patient repeatedly stuffed their ears with tissues and other objects. For the patient’s safety, the nurse initially removed the material filling the patient’s ears. However, because the patient was not doing anything that would cause physical or mental injury, the nurse ultimately decided to merely observe the patient and to not unduly restrict his movements. Later, the patient realized that stuffing their ears was not an effective way to suppress auditory hallucinations and ceased engaging in this behavior. Nurses thus recognize that patients can themselves realize the effectiveness, or otherwise, of their coping behaviors, such that nurses can modify a patient’s behavior without having to forcefully control them.


*Patients sometimes simply do not accept it. Even when I approach patients under the assumption that they appear to be calm, there are times when they are still not receptive to conversation. At such times, I do not try to forcefully draw out the patient’s thoughts and feelings or deepen the discussion; instead, I resume the conversation in a calm manner.*
(K)

##### Concept: Creating an Environment Where the Patient Can Remain Calm

Nurses work to help patients regain their composure by creating an environment with few stimuli that make them feel restless or agitated. The nurse tends to be aware of signs that the patient is becoming unstable. In such cases, nurses may avoid engaging in open observation during mealtimes and adjust environment so that objects in the room that may be irritating to the patient do not come into view. Nurses aim to minimize patient exposure to stimuli that symbolize negative experiences.


*The patient is quite vocal and restless before meal times. He suddenly starts saying “Food, food” and then says “I’m going home”. We are particular about meal times. Another time, I was calmly soaking up the sun and feeling relaxed, but then I suddenly became restless when I heard other patients and staff talking about food. Therefore, I avoided open observation during meal times and instead conducted open observation from 2 to 4 p.m.*
(I)


*When the patient is in the room, he talks about “someone coming in”, states that “the ceiling is falling down”, and becomes increasingly anxious. We wanted patients to feel at ease when they were in the room, so we decided to reverse the head and foot ends of the bed. By changing the position of the bed, ceiling vents and lighting may appear different when the patient looks up. Also, you may feel safer if you can’t see the entrance, if your head is facing the door. The patient’s line of sight is positioned so that the patient can see the balcony through the acrylic panel. Maybe the patient is not paying attention to the door. The patient likes where he is now.*
(M)

#### 3.3.3. Category: Promoting Self-Awareness of Maladaptive Behavior Patterns

Nurses help patients become aware of patterns of maladaptive behavior associated with situations that can promote mental instability. For example, nurses visually present signs and patterns to patients that predict an excited state, helping patients predict their own discomfort. Efforts are also made to make patients aware of areas in which they lack coping skills. Nurses repeatedly explain things to patients with intellectual disabilities, in plain language that they can understand, to help them realize that violence against others is a problem. Nurses also encourage patients to understand that agitation can interfere with daily life, such as by preventing bathing, which makes it difficult for nurses to assist patients.

##### Concept: Repeatedly Asking about Progress and Prospects

Nurses repeatedly talk to patients to help them understand their own responses and actions, such as the behaviors that led to seclusion or physical restraint, as well as the conditions that must be met to lift any coercive measures. However, even when nurses review their ideas with patients, they often find that although the patients may appear to be convinced immediately, they in fact often have difficulty in understanding or their understanding does not persist. Nurses repeatedly explain in advance the mental state that patients would have to be in for the use of a seclusion room, i.e., a state in which they cannot control their impulses. The nurse also informs the patient that the use of a seclusion room will likely reduce their agitation in the short term. Moreover, when a patient is watching another patient become violent, the nurse engages with the observing patient by asking him or her to reflect on their own violent behavior. Furthermore, if patients are being kept under quarantine for extended periods of time, nurses explain the prospects of their future release from seclusion. As mentioned above, nurses try to ensure patients have a sufficient understanding that protection rooms are used strategically to stabilize patients’ mental states.


*If a patient continues to behave in a particular way, such as getting angry at the nurse, acting violently in the room, or refusing to take his or her medication, I tell them that no matter how long they stall, they will not be discharged from the hospital because treatment will not progress. I am referring to things like not taking specific medicines properly, getting angry, or acting harshly toward medical professionals.*
(L)

##### Concept: Responses That Lead to Realistic Understanding

Nurses respond in a way that allows patients to come to terms with their own desires and anxieties, and to be able to live comfortably while receiving medical treatment. For example, nurses may remind patients of what they really need and discuss how to deploy the resources they currently have to avoid escalating demands. Nurses aim to approach patients in a calm and realistic manner. Patients may be affected by hallucinations or delusions and have difficulty understanding what nurses are saying to them. Even if the patient’s mental state is unstable, if the nurse takes an approach that accords with the patient’s wishes and goals, the patient will be able to divert his or her attention away from morbid experiences, which will stabilize their mental state. Nurses are aware that some patients are able to have conversations even if they remain in the seclusion room. Nurses place importance on establishing dialogue with a patient and strive to build a relationship where they can work together.


*Patients come to the nurse center to talk and make requests. One patient had the delusion that his clothes stank. The patient told the nurse, “My clothes stink again, so I want you to change them”, and “My body stinks, so I want you to look at my butt”. The nurse looked at the patient’s body and said, “You’re perfectly fine”, and the patient calmed down. The nurse looked over toward the patient and told him, “It’s okay”.*
(O)

#### 3.3.4. Category: Suggestions for Actionable Preventive Measures

A nurse may suggest coping skills that the patient can deploy to prevent any worsening of mental symptoms. We are working toward determining the factors that may cause a patient’s mental state to become unstable because of stimulation during open observation, and we are aiming to ensure that measures are in place to support patients so that they can spend their time in the ward with a sense of security.

##### Concept: Negotiating Means to Satisfy Goals and Wishes

Nurses negotiate practical means with patients to help fulfill their goals and wishes. Nurses work with patients to help them become more aware of what they can do (and how to do it) to make their time more comfortable and to satisfy their own needs. Patients tend to act according to their desires. For this reason, we discuss in detail the steps we take to help patients fulfil their aspirations and goals. The reason why such discussions are necessary is that patients’ goals may be perceived as being in the distant future, and it is difficult for them to think about how to approach their goals.


*The requests are always the same, such as wanting clothes or a watch. Once, we gave a patient a watch, but he broke it. For financial reasons, I can’t give out a watch every time. We thought about what we could do to placate the patient, so we asked him to make his own clock. The patient was satisfied and stopped banging on the wall. Together, nurses and patients consider how patients can meet their own needs.*
(Q)

##### Concept: Negotiating Acceptable Arrangements That Suit the Patient

The nurse helps the patient understand the reality of the situation and negotiates certain arrangements. Nurses anticipate that the patient’s demands will escalate, and they aim to “put the brakes on” the patient’s growing desires at an early stage. However, rather than just “putting on the brakes”, nurses also remind patients to continue using effective methods to achieve what they want. Nurses try to help patients control their desires and engage in goal-oriented behavior in their daily lives.


*There were times when I couldn’t get the patients to take a bath for about a month. If the patient was in a bad mood, he would get angry even if I asked him if he would like to take a bath. The patient was reluctant to take a bath or change his clothes. The patient did not feel that delusions were involved. I didn’t feel like there was any reason for his reluctance. I get the impression that it doesn’t matter if you don’t enter. I suggested some minimum goals to him, such as, “Let’s change our clothes at least once a week”.*
(J)

##### Concept: Creating a Place Where Patients Can Be Motivated to Be Active

By performing activities that patients are interested in during open observation, nurses make adjustments so that patients can carry out activities that will motivate them to work toward their own goals. Nurses believe that active engagement allows patients to work on long-term goals. Nurses also encourage patients to engage in activities that require their existing abilities to be exercised during open observation. This can be viewed as an effort to leverage meaningful activities to promote functional recovery in the patient.


*The patient often reads newspapers and watches baseball games on TV. When a patient does something that they normally do when they are feeling well, I say to them, “You’re reading the newspaper properly today”. Other activities include attending occupational therapy and riding a stationary bike. When I see someone riding an exercise bike, I say to them, “I see you’re riding your bike today”. I think a good indicator of how a patient is doing is seeing that they are motivated to ride the bike.*
(M)

##### Concept: Self-Choice in Activities and Symptom Management

As far as possible, nurses create opportunities for patients to make their own choices with respect to activities and symptom management. Nurses recognize interactions that encourage patients to become aware of how what they want can actually lead to the worsening of their mental symptoms. Recognizing that the more they try to control a patient’s behavior the more stress it causes for the patient, nurses emphasize letting patients decide for themselves as far as possible. Nurses feel that the more patients decide for themselves, the more likely it is that choices with positive consequences will be remembered. Furthermore, when a patient is mentally unwell, the decision is made to allow them to request assistance from the nurse on their own, and to choose their own coping skills. Additionally, when extending the open observation time to encompass patient seclusion, nurses confirm the patient’s wishes in advance and consult with the doctor.


*Patients can tap on the nursing center window and say, “I need some help”. However, patients may not give us time to prepare for the situation. The patient sends out an “SOS”, and I ask, “So what do we do? Sometimes the patient says, “But I don’t want to drink”, and sometimes they say, “Okay, I’ll drink”. How the patient behaves is left up to them. Of course I won’t force it. However, if a patient is feeling unwell for a long time or is in a state of agitation, we may recommend coercive measures.*
(B)


*Patients can walk around the hospital with the assistance of a nurse. I asked a patient, “What do you want to do?” and the patient said that he wanted to sit in a chair. Patients are often relaxing in a chair in front of the TV, and we can confirm that their stated intentions were genuine. Additionally, even when I am observing patients in open conditions, they may want to go back to their rooms, so I respect their wishes and allow them to be where they want to be.*
(I)

## 4. Discussion

Characteristic features of the SDM process uncovered in this study included preventive measures taken through the initiative of nurses, as well as respecting the choices of patients by presenting then with various types of preventive measures. These two contrasting aspects are influenced by the patient’s stimulus tolerance. To aid recovery from SMI, promoting autonomously motivated behavior and creating opportunities for self-determination are encouraged [15]. In addition, although nurses must respond in a limited manner to patients’ inappropriate behavior, they nevertheless strive to engage in mutually beneficial communication and seek to promote collaboration with patients with SMI through daily activities [13]. Even in research conducted from the patient’s perspective, statements expressing the desire for patients to maintain their autonomy as an aspect of the nursing intervention were reported [19]. Efforts to enhance patients’ self-care and autonomy are important in psychiatric nursing.

However, because patients with SMI have difficulty managing their symptoms on their own, nurses need to take the lead in detecting mental illness and guiding patients in the use of coping skills. Psychiatric nursing for patients with coexisting mental illness and intellectual disability needs to reduce stimulation to ensure the safety of everyone in the treatment environment, and the difficulty in identifying the factors underlying the patient’s behavior should be acknowledged [20]. Some patients who undergo long-term coercive measures have a combination of schizophrenia and intellectual disability, which is complicated to manage. Nurses evaluate the patient’s reactions to environmental stimuli and then determine whether they (or the patient) should take the lead in terms of decision-making.

Nurses work toward helping the patient become aware of their own patterns of agitation. Patients with SMI have difficulty with self-reflection and are less likely to recognize that their own maladaptive behaviors lead to their segregation. Previous research has shown that this practice can improve treatment outcomes and promote collaboration [13]. Additionally, improvements in patients’ self-management abilities and therapeutic relationships have reportedly been obtained through the practice of SDM [21]. The underlying concept of this study is that nurses can support patients so that their understanding of the relationship between behavior and treatment is deepened, and to provide them with relief from anxiety arising in association with their pathological experiences. It has been reported that some healthcare professionals allow patients to deal with negative emotions on their own [22]. The approach of “repeatedly asking about progress and prospects” helps patients realize that negative emotions are linked to aggression toward themselves and others. This approach can promote awareness of negative emotions, which is a prerequisite for dealing with them. In other words, if patients can better understand the triggers and factors that increase their risk of exhibiting maladaptive behaviors, they may have a better chance of avoiding seclusion or physical restraint.

The nurses in this study reported negotiating with patients to help them fulfill their goals and wishes, though this depends on the extent to which the patient’s requests were realistic. It has been pointed out that personal goal setting and a sense of shared responsibility are important aspects of SDM with SMI patients [13]. Furthermore, shared responsibility contributes to a reduction in patient violence and promotes consistency in healthcare workers’ attitudes [23]. Patients’ demands tend to escalate, making it difficult for nurses to respond adequately. Discontinuity, insufficient time, and coercion have been reported as unhelpful aspects of therapeutic relationships from the perspective of SMI patients [24]. Nurses may be asked to make very complex adjustments to meet patient needs while also avoiding any escalation of demands.

Furthermore, the nurses in this study aimed to engage in decision-making that respected patients’ wishes as much as possible, working toward fulfilling each patient’s goals and wishes and providing space for ambitious activities. It was reported that motivation and performance were significantly associated in people with schizophrenia, regardless of age or the severity of the psychopathology [25]. SDM takes into account the patient’s cultural values, beliefs, and preferences, allowing a collaborative relationship to be built with the patient [26] and supporting the active participation of the patient in their treatment [21]. In this study, an environment was created in which patients’ goals and wishes, as well as their preferred activities, were promoted; the environment was made as comfortable as possible to encourage patients’ self-determination, even though space was restricted. Building relationships on the basis of SDM promotes patient empowerment and personal recovery [27]. A study investigating the management of patient aggression reported that patients sought individual attention from professionals as their aggression increased [22]. This study also identified an approach in which nurses kept the patient’s goals and wishes in the foreground when making suggestions for actionable, preventive measures, and they sought to identify acceptable lifestyle behaviors for the patient. Thus, psychiatric nurses can respond in a personalized manner to meet patients’ needs.

This study has some limitations. First, only qualitative analysis of the nursing process, as it relates to SDM from the nurse’s perspective, was performed, and data on the background information of patients undergoing long-term coercive measures were not collected. The side effects of drug therapies in patients with SMI, and their impact on patients’ decision-making and quality of life, remain unclear. Furthermore, because we did not investigate the extent to which patients with SMI understand the content or effects of their treatment, the degree to which patients accept and participate in their treatment and nursing care remains unclear. Moreover, this study viewed the collaborative decision-making process for patients with SMI receiving treatment involving long-term coercive measures from the nurses’ perspective. Furthermore, the results pertain only to the nursing process as it relates to SDM aimed at minimizing the need for psychiatric nurses to implement coercive measures for patients with SMI, and at this stage, the findings can be considered theoretical. It is necessary to evaluate the applicability of the generated theory to other similar clinical settings and to conduct research to further modify and optimize the theory. It should be noted that the effects of SDM-based interventions on clinical and personal recovery, the contribution of SDM to the minimization of coercive measures, and the subjective experiences of patients were not evaluated. Future studies need to evaluate these topics through further interventions based on SDM between patients and nurses.

## 5. Conclusions

In the SDM process, which involves enhancing the patient’s coping skills and living environment and minimizing the use of seclusion and physical restraint, “sensing the response to triggering stimuli” and “nurse-led preventive measures” compensated for any lack of coping skills of the patient. Moreover, nurses strived to create opportunities for repeated learning about the triggers of patients’ maladaptive behaviors by “promoting self-awareness of maladaptive behavior patterns” and “suggestions for actionable preventive measures”.

Nurses reportedly evaluate patients’ reactions to environmental stimuli and then determine whether they or the patient should take the lead in decision-making. Even for patients receiving treatment including coercive measures, nurses seek to create an environment that is as comfortable as possible to promote patients’ self-determination. Building relationships on the basis of SDM may lead to patient empowerment and improved personal recovery.

## Figures and Tables

**Figure 1 healthcare-12-00967-f001:**
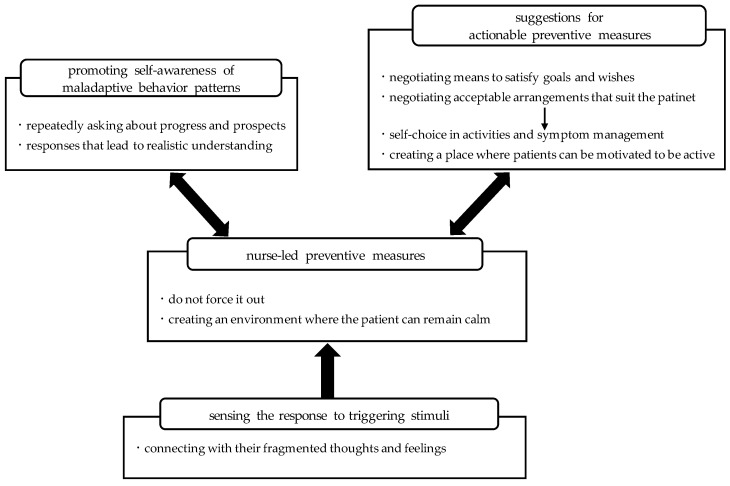
The nursing process for shared decision-making for patients with severe mental illness receiving treatment involving long-term coercive measures.

**Table 1 healthcare-12-00967-t001:** Participants’ demographic information and patient characteristics as described by the participants.

Participant ID	Years of Nursing Experience	Years of Psychiatric Nursing Experience	Diagnosis of Patients as Described by Participants	Reasons for Coersive Measures	Types of Coersive Measures
A	7	4	Schizophrenia, intellectual developmental disorder	Self-harm	Seclusion, physical restraint
B	8	8	Schizophrenia	Agitation, restlessness, self-harm, harm to others	Seclusion, physical restraint
C	10	10	Schizophrenia, intellectual developmental disorder	Agitation, restlessness, harm to others, nuisance	Seclusion
D	7	2	Schizophrenia	Agitation, restlessness, self-harm, harm to others	Seclusion, physical restraint
E	20	14	Schizophrenia	Harm to others	Seclusion
F	28	28	Schizophrenia, intellectual developmental disorder	Harm to others, polydipsia	Seclusion
G	25	15	Schizophrenia, intellectual developmental disorder	Agitation, restlessness, harm to others	Seclusion
H	19	19	Schizophrenia, intellectual developmental disorder	Self-harm, harm to others	Seclusion, physical restraint
I	10	3	Schizophrenia	Agitation, restlessness	Seclusion
J	23	20	Schizophrenia	Self-harm, harm to others	Seclusion, physical restraint
K	15	10	Schizophrenia	Self-harm	Seclusion, physical restraint
L	10	6	Schizophrenia	Self-harm	Seclusion, physical restraint
M	24	11	Schizophrenia	Self-harm	Seclusion, physical restraint
N	26	19	Schizophrenia, intellectual developmental disorder	Harm to others, nuisance	Seclusion
O	10	6	Schizophrenia	Harm to others, nuisance	Seclusion
P	11	11	Schizophrenia	Harm to others	Seclusion
Q	13	7	Schizophrenia	Harm to others	Seclusion

## Data Availability

The qualitative dataset and transcriptions of narratives are not publicly available because of ethical restrictions and privacy issues.

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
