# Peer review of "A Nursing Process for Shared Decision-Making for Patients with Severe Mental Illness Receiving Treatment Involving Long-Term Coercive Measures: A Modified Grounded Theory Approach"

_healthcare, 2024, doi:10.3390/healthcare12100967_

Round 1

Reviewer 1 Report

Comments and Suggestions for Authors

Dear author,

 The topic chosen for your analysis is a very interesting one and deserves to be studied in more detail. The reason why the duration of coercive measures such as segregation and physical restraint tends to be longer in Japan also deserves attention. Every effort must be made for SMI patients to improve their quality of life, and this aspect concerns not only the nurses, but the entire medical staff, the family, the society in which they live and the laws of the respective country. Indeed, SDM may be applied as a mean to build recovery-oriented therapeutic relationships with patients with SMI.

I have some comments and recommendations:

-Line 88-91 – Participants: I think that the experience of a nurse who worked for only one month with patients with SMI who require physical isolation or restraint measures is limited. I would suggest studying only nurses with at least two years of experience in a psychiatric clinic and at least 6 months of experience with serious cases.

-Line 125: Also, only 17 nurses were recruited for the study, which is a very small number to draw significant conclusion or to generalize the results.

Discussions: A limitation of the study section is needed, and also more comparative discussions with other similar studies. 

Reviewer 2 Report

Comments and Suggestions for Authors

Does the introduction provide sufficient background and include all relevant references?

A simple explanation as to M-GTA and the identified concepts would have presented a stronger introduction. The title is misleading as 'SDM between nurse and patients', however, the interviews involved nurses, not patients. (line 96)

The author identified a conceptual analysis for the study. There are several concepts represented in figure 1, however, the categories that go with specific concepts are not linked, making this hard to follow for the reading audience. While methods are mentioned, as stated, the flow diagram with analytical themes and focus are hard to follow.

Is the research design appropriate?

No design other than the M-GTA was identified. Offering a simple definition as to M-GTA (is a qualitative research methodology that attempts to unravel the meanings of people's interactions, social actions, and experiences. In other words, these explanations are grounded in the participants' own interpretations or explanations) would have helped in understanding the approach the author presented and why. The author lists for basic principles for M-GTA, but lacks expanding these (lines 80-85). Further development of the grounded theory approach is necessary.

Are the methods adequately described?

I have many questions as to this type of treatment option for severe mental illness. I also realize these may be outside the scope of the study (line 344-348) and could be offered as a recommendation/s for future research.

-        Are the patients able to comprehend what this coercive treatment means to them? Many individuals in either 2-point or 4-point restraints require sedation just to protect them from self-harm as they  attempt to manipulate the leathers and release themselves from this type of physical control. How would the author address this type of control, as ultimately this is the treatment: control of one individual over another.

-        Are chemical restraints considered in conjunction with physical or seclusion?

-        Are the patients participating in focus groups, individual therapy sessions with medical AND nursing professionals?

-        How effective is this type of treatment over time? Have any studies been conducted retrospectively to evaluate the outcomes?

-        What population is served? All classes including homeless, single parents, other vulnerable populations? The elderly?

Line 332-334: I am not sure what the author is attempting to convey. Please review and revise for clarity.

Are the conclusions supported by the results?

SMI includes schizophrenia, bipolar, OCD, panic episodes, PTSD, and other major mental health diagnoses. As such, are the patients that are admitted for inpatient coercive treatment cognitive enough to make a self-determination in their care. The author discussed and concluded that this vulnerable population can develop coping skills, recognize when their stress or behavior is escalating into a maladaptive state.

Round 2

Reviewer 1 Report

Comments and Suggestions for Authors

Dear Authors,

Thank you for responding to my comments and recommendations. I think your manuscript is improved and can be published.

Kind regards.

Author Response

Dear Reviewer

Thank you very much for your valuable advice to improve the quality of my research paper.

Best regards.

Reviewer 2 Report

Comments and Suggestions for Authors

The improvements to each comment from the first review have enhanced the manuscript's readability and intent for the reading audience. Great improvement.

One comment concerns Table 1, which combines the participants' demographic information and the patient characteristics as described by the participants. A suggestion is a vertical line separating the two (between years of nursing experience and the diagnosis of patients)...

Clear labeling and detailed explanations are crucial for the audience's understanding of the nursing process. I think your role in providing these is crucial. I still need help understanding the flow chart for Figure 1. The figure is labeled as nursing process . . .I would like to see a clearer labeling of the nursing process and the interventions that the author has listed.

For example, in the nursing process (numbered), the nurse senses the response of patients to triggering stimuli. Intervention: nurse will facilitate the patient in connecting with their fragmented thoughts and feelings by . . .(how will they do this)

For example, the nursing process: nurse-led preventative measures (this needs a more definitive description; what preventative measures?). Intervention: do not force it out (more detail, even though I know what you have said for this statement; it needs more detail here). Intervention: creating an environment where the patient can remain calm: how?

Think of your reading audience. How would you explain this to a nurse who is not familiar with psychiatric coercive restraints? Someone who has no psychiatric background.

If you list an intervention, could you include how you will accomplish what you describe? How will you determine if the patient has a realistic understanding of what????

How cognitively compromised are the patients treated in this type of inpatient unit? Are they aware of what treatments are administered and how they will affect them? How can you determine their cognitive state and ability to self-determine in their collaborative care?

I know you have mentioned some of these items in the discussion, review and revise for emphasis if needed.

Comments on the Quality of English Language

There are a few places where grammar will need repair. Run the manuscript through a grammar checker.
